# R2D2: Reuse & Reduce via Dynamic Weight Diffusion for Training Efficient NLP Models

## Abstract

We propose R2D2 layers, a new neural block for training efficient NLP models. Our proposed method is characterized by a dynamic weight diffusion mechanism which learns to reuse and reduce parameters in the conventional transformation layer, commonly found in popular Transformer/LSTMs models. Our method is inspired by recent Quaternion methods which share parameters via the Hamilton product. This can be interpreted as a neural and learned approximation of the Hamilton product which imbues our method with increased flexibility and expressiveness, i.e., we are no longer restricted by the 4D nature of Quaternion weight sharing. We conduct extensive experiments in the NLP domain, showing that R2D2 (i) enables a parameter savings of up to $2\times$ to $16\times$ with minimal degradation of performance and (ii) outperforms other parameter savings alternative such as low-rank factorization and Quaternion methods.

## 1 Introduction

The transformation layer is one of the most ubiquitous and dominant component in existing and current deep learning literature (Goodfellow et al., 2016). Its pervasiveness cannot be understated, given its centrality to many core building blocks in neural network research. Given widespread adoptions of FC layers, e.g., within Transformer (Vaswani et al., 2017) models and recurrent (Hochreiter & Schmidhuber, 1997) networks, a further reduction in parameter complexity and memory footprint could be extremely useful to many real world applications.

This paper proposes a new neural component for reusing and reducing parameter costs of the transformation layer (i.e., memory footprint). Concretely, our proposed R2D2 layer is characterized by a new dynamic weight diffusion mechanism. The central idea is that we start off with a core set of parameters (weights) and learn dynamic 'diffusion' of these weight partitions to construct the final transformation weight $W$. The key novelty lies in our method of incorporating soft weight reuse, i.e., taking inspiration from multiplication in hypercomplex spaces.

Recent work in Quaternion spaces and Hamilton products have demonstrated reasonable success (Parcollet et al., 2018; 2019; Tay et al., 2019). The Hamilton product, which multiples two Quaternions by fusing latent inter-component representations, enables a four time parameter saving as compared to the real-valued adaptation. Unfortunately, Hamilton products operate on 4D hypercomplex numbers, which limits its expressiveness and utility. To this end, our proposed R2D2 layer can be viewed as a neural approximation of the Hamilton product, learning the permutation of the inter-latent component interactions in a soft differentiable fashion. Moreover, our method can operate on any arbitrary $n$, aside from only on $4D$ Quaternion representations, facilitating up to $n$ times parameter savings.

To demonstrate applicability, we equip two well-established models (LSTMs and Transformers) with R2D2 layers. We conduct extensive experiments on flagship benchmarks, *i.e.*, neural machine translation for Transformers and natural language inference for LSTMs. Additionally, we include further validation on text style transfer and subject-verb agreement tasks. All in all, we find that R2D2 generally enables up parameter savings up to approximately $1/16 \sim 1/2$ with minimal degradation in performance. Moreover, our proposed method also enables slight speed-up in terms of inference (decoding speed).

## 2 OUR PROPOSED METHOD

This section introduces our proposed R2D2 components.

### 2.1 VANILLA FC LAYERS

Before we delve into our proposed method, the standard FC transform layer is defined as:

$$\mathbf{y} = \text{FC}(\mathbf{x}) = \mathbf{W}\mathbf{x} + \boldsymbol{b}. \tag{2.1}$$

The FC layer (equation 2.1) is fundamental to many modern and traditional neural network architectures. Owing to its ubiquity and widespread usage, an efficient reduction of parameters in the FC layer may result in massive parameter savings, especially for models that heavily use transformation equation 2.1.

**Remark 2.1** *The degree of freedom for the weight parameters* $\mathbf{W}$ *in equation 2.1 is* $kd$. *Since* $\mathbf{W}$ *dominates parameterization, the parameter size of the FC layer equation 2.1 is* $\mathcal{O}(kd)$.

In the following, we describe the proposed R2D2 layer and its relationships with matrix multiplication in real space and Hamilton product in hypercomplex space.

### 2.2 REUSE AND REDUCE WITH DYNAMIC WEIGHT DIFFUSION (R2D2)

Preserving the same notation from the FC layer equation 2.1, a R2D2 layer transforms an input $\mathbf{x} \in \mathbb{R}^d$ into an output $\mathbf{y} \in \mathbb{R}^k$ with $\mathbf{H} \in \mathbb{R}^{k \times d}$ and $\boldsymbol{b} \in \mathbb{R}^k$:

$$\mathbf{y} = \text{R2D2}\,(\mathbf{x}) = \mathbf{H}\mathbf{x} + \boldsymbol{b}. \tag{2.2}$$

The distinct difference in R2D2 is that we construct the parameter $\mathbf{H}$ in equation 2.2 with a reduced degree of freedom in order to reduce the parameter size. This is achieved by our Dynamic Weight Diffusion method described as follows:

#### 2.2.1 DYNAMIC WEIGHT DIFFUSION

This section describes our proposed Dynamic Weight Diffusion mechanism. The central idea is to operate on partitioned[1] weight blocks and learn a dynamic diffusion of weights. There are two key parameter blocks $\mathbf{A}$ and $\mathbf{S}$ which are central to our approach. Intuitively, $\mathbf{A} \in \mathbb{R}^{n \times n \times n}$ controls the weight diffusion process and learns the soft interaction between $\mathbf{S}$ partitions. Here, $n$ is a user defined hyperparameter.

Suppose that both $d$ and $k$ are divisible by $n \in \mathbb{Z}_{>0}$. For $i = 1, \ldots, n$ and $j = 1, \ldots, \frac{d}{n}$, denote by each partitioned parameter block $\mathbf{S}_j \in \mathbb{R}^{n \times \frac{k}{n}}$. $\mathbf{A}_i \in \mathbb{R}^{n \times n}$ is the weight diffusion matrix assigned to each of $n$ blocks. The parameter $\mathbf{H}$ in equation 2.2 is constructed by column-wise concatenation (;):

$$\mathbf{H} = [s(\mathbf{A}_1); s(\mathbf{A}_2); \ldots; s(\mathbf{A}_n)], \tag{2.3}$$

where each segment $s(\mathbf{A}_i)$ is also formed by column-wise concatenation:

$$s(\mathbf{A}_i) = [\psi(\mathbf{A}_i\mathbf{S}_1); \psi(\mathbf{A}_i\mathbf{S}_2); \ldots; \psi(\mathbf{A}_i\mathbf{S}_{\frac{d}{n}})]. \tag{2.4}$$

In equation 2.4, function $\psi : \mathbb{R}^{p \times q} \to \mathbb{R}^{pq}$, where $\psi(\mathbf{X})$ flattens the matrix $\mathbf{X} \in \mathbb{R}^{p \times q}$ by concatenating each row of $\mathbf{X}$ then transposes the concatenated row vector into a column vector of dimension $pq$. It is easy to see that, $\psi(\mathbf{A}_i\mathbf{S}_j) \in \mathbb{R}^k$, $s(\mathbf{A}_i) \in \mathbb{R}^{k \times \frac{d}{n}}$, thus $\mathbf{H} \in \mathbb{R}^{k \times d}$.

---

[1]Here, the partitioning (or splitting) of weights into different components is analogous to complexification, i.e., splitting real vectors into real and imaginary components.

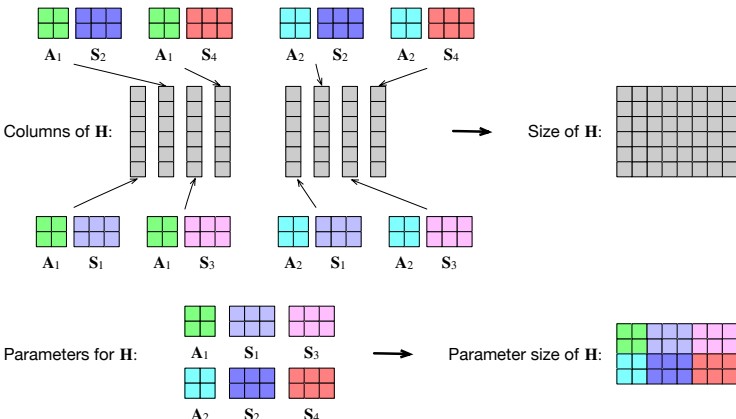

Figure 1: Illustration of our proposed Dynamic Weight Diffusion. Reusing partitioned parameter blocks $\mathbf{A}_i$ ($i = 1, 2$) and $\mathbf{S}_j$ ($j = 1, \ldots, 4$) leads to a reduction in the parameter size of $\mathbf{H}$ ($n = 2, d = 8, k = 6$). Best viewed in color.

As illustrated in Figure 1, the reuse of partitioned parameter blocks $\mathbf{S}_1, \ldots, \mathbf{S}_{\frac{d}{n}}$ in the $n$ segments $s(\mathbf{A}_i)$ ($i = 1, \ldots, n$) in equation 2.3 and the reuse of the partitioned parameter block $\mathbf{A}_i$ in each individual $s(\mathbf{A}_i)$ segment in equation 2.4 may reduce the degree of freedom for $\mathbf{H}$.

**Remark 2.2** *It is partitioned parameter blocks $\mathbf{A}_i$ ($i = 1, \ldots, n$) and $\mathbf{S}_j$ ($j = 1, \ldots, \frac{d}{n}$) that determine the degree of freedom for $\mathbf{H}$, which is $\frac{kd}{n} + n^3$. Since $\mathbf{H}$ dominates parameterization, the parameter size of the R2D2 in equation 2.2 is $\mathcal{O}(\frac{kd}{n})$, where $kd \gtrsim n^4$ is assumed: this condition is mild for real-world problems, such as in our experiments (e.g., $d = 512$, $k = 2048$, $n = 2, 4, 8, 16$). Thus, for the same input and output sizes, the parameterization cost of a R2D2 layer is approximately $\frac{1}{n}$ of that of an FC layer under mild assumptions.*

## 2.3 RELATIONSHIP WITH MATRIX MULTIPLICATION IN REAL SPACE

The R2D2 layer subsumes matrix multiplication in real space. Thus, it is a generalization of the FC layer via hyperparameter $n$. To explain, referring to equation 2.2, when $n = 1$, $\mathbf{H} = a\mathbf{W}$, where scalar $a$ is the single element of the $1 \times 1$ matrix $\mathbf{A}_1$ and elements of $\mathbf{W} \in \mathbb{R}^{k \times d}$ come from the concatenation of $\mathbf{S}_1, \ldots, \mathbf{S}_d \in \mathbb{R}^{1 \times k}$. Since learning $a$ and $\mathbf{W}$ separately is equivalent to learning their multiplication jointly, scalar $a$ can be dropped, which is learning the single weight matrix in an FC layer. Therefore, a R2D2 layer is degenerated to an FC layer when $n = 1$, where there is no parameter saving.

## 2.4 RELATIONSHIP WITH HAMILTON PRODUCT IN HYPERCOMPLEX SPACE

Next, we explore the relationship between the R2D2 layer and Hamilton product in hypercomplex space.

For background, in hypercomplex space, inputs are represented by multiple imaginary components. For example, a Quarternion has 1 real component and 3 imaginary components, while an Octonion has 1 real component and 7 imaginary components. For the sake of illustration, let us take a Quarternion $Q$ as an example, which is a 4-dimensional hypercomplex number

$$Q = Q_r + Q_x \mathbf{i} + Q_y \mathbf{j} + Q_z \mathbf{k}, \tag{2.5}$$

where $Q_r, Q_x, Q_y, Q_z$ are real numbers, $\mathbf{ijk} = \mathbf{i}^2 = \mathbf{j}^2 = \mathbf{k}^2 = -1, \mathbf{ij} = \mathbf{k}, \mathbf{jk} = \mathbf{i}, \mathbf{ki} = \mathbf{j}, \mathbf{ji} = -\mathbf{k}, \mathbf{kj} = -\mathbf{i}, \mathbf{ik} = -\mathbf{j}$. The Hamilton product, which represents the multiplication of two Quaternions $S = S_r + S_x \mathbf{i} + S_y \mathbf{j} + S_z \mathbf{k}$ and $Q$ equation 2.5, is defined as

$$\begin{bmatrix} S_r & -S_x & -S_y & -S_z \\ S_x & S_r & -S_z & S_y \\ S_y & S_z & S_r & -S_x \\ S_z & -S_y & S_x & S_r \end{bmatrix} \begin{bmatrix} Q_r \\ Q_x \\ Q_y \\ Q_z \end{bmatrix}, \tag{2.6}$$

where the 4 output elements are the real values for the Quaternion unit basis $[1, \mathbf{i}, \mathbf{j}, \mathbf{k}]^\top$. Denoting that $\mathbf{s} = [S_r, S_x, S_y, S_z]^\top$, equation 2.6 can be rewritten as

$$\left( \begin{bmatrix} 1 & 0 & 0 & 0 \\ 0 & 1 & 0 & 0 \\ 0 & 0 & 1 & 0 \\ 0 & 0 & 0 & 1 \end{bmatrix} \mathbf{s}; \begin{bmatrix} 0 & -1 & 0 & 0 \\ 1 & 0 & 0 & 0 \\ 0 & 0 & 0 & 1 \\ 0 & 0 & -1 & 0 \end{bmatrix} \mathbf{s}; \begin{bmatrix} 0 & 0 & -1 & 0 \\ 0 & 0 & 0 & -1 \\ 1 & 0 & 0 & 0 \\ 0 & 1 & 0 & 0 \end{bmatrix} \mathbf{s}; \begin{bmatrix} 0 & 0 & 0 & -1 \\ 0 & 0 & 1 & 0 \\ 0 & -1 & 0 & 0 \\ 1 & 0 & 0 & 0 \end{bmatrix} \mathbf{s} \right) \begin{bmatrix} Q_r \\ Q_x \\ Q_y \\ Q_z \end{bmatrix}. \tag{2.7}$$

It can be observed that, when $n = k = d = 4$, the R2D2 layer may perform Hamilton Product of Quarternions. Specifically, partitioned parameter blocks $\mathbf{A}_1, \ldots, \mathbf{A}_4$ in equation 2.3 parameterize the 4 permutation blocks (matrices composed of $-1, 0, 1$ in equation 2.7) that reflect the rules of Hamilton product, while $\mathbf{S}_1$ in equation 2.4 are $\mathbf{s}$ in equation 2.7, and the layer input $\mathbf{x}$ in equation 2.2 is $[Q_r, Q_x, Q_y, Q_z]^\top$ in equation 2.7. Likewise, Hamilton product of Octonions or Sedenions in hypercomplex space may also be performed by the R2D2 layer when $n, k, d$ are equally set to 8 or 16.

### 2.5 GENERALIZING HYPERCOMPLEX MULTIPLICATION WITH NEURAL APPROXIMATION

In fact, weight reuse by component-wise partitioning in Quaternion space has demonstrated reasonable success (Parcollet et al., 2018; Zhu et al., 2018; Parcollet et al., 2019; Tay et al., 2019). However, one key problem is that Hypercomplex algebra cannot be generalized to arbitrary $n$ values simply because algebraic axioms cannot hold when $n$ takes values apart from power of 2. At this point, it is easy to see that 5-dimensional complex numbers cannot exist due to a clear inability to multiply two 5-dimensional complex numbers. Within the context of hypercomplex space, specialized multiplication rules, such as Hamilton product, have to be devised and encoded in the network as a fixed inductive bias. In sharp contrast, the R2D2 layer learns such specialized multiplication rules from data, as manifested in partitioned parameter blocks $\mathbf{A}_i$ $(i = 1, \ldots, n)$ in equation 2.3. Thus, R2D2 subsumes and is degenerated to hypercomplex multiplication when such partitioned parameter blocks are set to reflect those predefined multiplication rules in hypercomplex space. Moreover, the proposed R2D2 layer can be seen as a trainable and parameterized form of $n$-dimensional hypercomplex multiplication, where $n$ can be values other than power of 2.

To sum up, the R2D2 layer reduces parameterization cost by reusing partitioned parameter blocks. It also offers a way to bridging multiplication between real space and hypercomplex space.

## 3 EFFICIENT NEURAL NLP MODELS WITH R2D2 LAYERS

To demonstrate the applicability of the R2D2 layers, we develop two popular neural network models, LSTM and Transformer, which are able to benefit from R2D2 layers.

### 3.1 R2D2-LSTMS

Recurrent neural networks such as LSTMs are gated recurrent networks where gating functions are parameterized by linear transformations. We introduce the our efficient LSTM (R2D2-LSTM), which replace such linear transformations in the LSTM with the R2D2 layers:

$$\mathbf{y}_t = \text{R2D2}\left(\mathbf{x}_t\right) + \text{R2D2}\left(\mathbf{h}_{t-1}\right) + \boldsymbol{b}$$
$$\mathbf{f}_t, \mathbf{i}_t, \mathbf{o}_t, \mathbf{x}'_t = \phi(\mathbf{y}_t)$$
$$\mathbf{c}_t = \sigma_s(\mathbf{f}_t)\,\mathbf{c}_{t-1} + \sigma_s(\mathbf{i}_t)\,\sigma_t(\mathbf{x}'_t)$$
$$\mathbf{h}_t = \mathbf{o}_t \odot \mathbf{c}_t$$

where $\sigma_s$ is the sigmoid activation function and $\sigma_t$ is the tanh activation function, $\phi : \mathbb{R}^{1 \times d} \to \mathbb{R}^{4 \times \frac{d}{4}}$ is a four way split on the last dimension, $c_t, h_t$ are the cell state and hidden state of the R2D2 -LSTM unit at time step $t$.

## 3.2 R2D2 TRANSFORMERS

Transformers (Vaswani et al., 2017), the state-of-the-art model for sequence transduction task, is a stacked neural network architecture that aggressively exploits linear transformations. Each self-attention layer comprises of three $\mathbf{Q}, \mathbf{K}, \mathbf{V}$ transformations, along with $N_h$ heads per self-attention layer. Each layer also comprises of a two layered FC layer with nonlinearities. A large majority of the Transformer parameters stem from the FC layers. In R2D2 Transformers, we replace all FC layers with R2D2 layers. The single-headed self-attention module is rewritten as:

$$\mathbf{Q}, \mathbf{K}, \mathbf{V} = \Phi(\text{R2D2}(\mathbf{X})) \quad , \quad \mathbf{A} = \text{softmax}(\frac{\mathbf{Q}\mathbf{K}^\top}{\sqrt{d_k}})\mathbf{V},$$

where $d_k$ is the key dimension, $\Phi : \mathbb{R}^{1 \times d} \to \mathbb{R}^{3 \times \frac{d}{3}}$ is a three way split on the last dimension, $\mathbf{X}$ is the input sequence, and $A$ is the self-attentive representation. For multi-headed, using R2D2 layers also enables weight sharing not only among $\mathbf{Q}, \mathbf{K}, \mathbf{V}$ but also among heads. attention, the transformation of multiple heads are also projected with R2D2 layers:

$$\mathbf{X} = \text{R2D2}([\mathbf{H}_1; \ldots; \mathbf{H}_{N_h}]),$$

where $N_h$ is the number of heads and $(;)$ is the column-wise concatenation. Finally, the position-wise FC layer is now defined as

$$\mathbf{Y} = \text{R2D2}(\text{ReLU}(\text{R2D2}(\mathbf{X}))),$$

which transforms $\mathbf{X}$ with 2 R2D2 layers.

## 4 EXPERIMENTS

This section reports the experimental results of R2D2-LSTMs and R2D2 Transformers that are equipped with R2D2 layers. Overall, we conduct 4 main experiments in which all experimental settings and hyperparameters (e.g., parameter initialization) remain consistent with the vanilla version. We evaluate R2D2 LSTMs on natural language inference (NLI) and compare them with standard LSTMs. Next, we evaluate R2D2 Transformers and compare them with standard Transformers on neural machine translation (NMT), text style transfer, and subject verb agreement (SVA) tasks.

### 4.1 NATURAL LANGUAGE INFERENCE

The task of natural language inference is to determine if two sequences entail or contradict with each other (MacCartney, 2009). NLI is a fundamental task pertaining to language understanding. To this end, they serve as a suitable benchmark for evaluating recurrent models.

Table 1: Experimental results on natural language inference. Saving cost considers only the cell unit (other parts of architecture remains constant). R2D2-LSTM reduces the parameter costs of the standard LSTM model and improves performance on 4 out of 5 datasets.

| Model | #Params | MNLI | QNLI | SNLI | DNLI | SciTail |
|---|---|---|---|---|---|---|
| LSTM | 721K | **71.82** / 71.89 | 84.44 | 84.18 | 85.16 | 74.36 |
| Quaternion LSTM | 180K (-75.0%) | 71.57 / **72.19** | **84.73** | 84.21 | 86.45 | 75.58 |
| R2D2-LSTM ($n = 2$) | 361K (-49.9%) | **71.82** / 72.08 | 84.39 | 84.38 | 85.77 | 77.47 |
| R2D2 LSTM ($n = 5$) | 146K (-79.7%) | 71.80 / 71.77 | 83.87 | **84.58** | **86.47** | 74.64 |
| R2D2-LSTM ($n = 10$) | 81K (-88.7%) | 71.59 / 71.59 | 84.25 | 84.40 | 86.21 | **77.84** |

**Datasets and Setup** We run experiments on 5 NLI datasets, *i.e.*, (1) SNLI (Bowman et al., 2015), (2) MultiNLI (Williams et al., 2017), (3) Dialogue NLI (Welleck et al., 2018), (4) QNLI (Quora) (Wang et al., 2017), and (5) Scitail (Science Entailment) (Khot et al., 2018). We implement 300-dimensional unidirectional encoders with shared parameters for both premise/hypothesis. We take

the concatenation of max and mean pooled representations as input to a two-layered 300-dimensional MLP for prediction. Our model is trained with the Adam with a learning rate of $0.0004$ with a batch size of 256. Word embeddings are initialized with GloVe (Pennington et al., 2014) and are fixed. No cross sentence attention (Parikh et al., 2016) is used, mainly to observe the effectiveness of standalone encoders. For R2D2-LSTM, we use $n = \{2, 5, 10\}$ and report the results accordingly. In this task, since word embeddings are 300-dimensional, we select multiples of 5 instead of 4 for simplicity and ease of divisibility.

**Experimental Results**  Table 1 reports the results on all the 5 natural language inference datasets. All in all, results are extremely encouraging, showing that the R2D2 layer can not only reduce the parameterization cost but also improve performance (4 out of 5 datasets show reasonable improvement). On QNLI (the only exception), the performance drop is marginal ($< 1\%$), which is decent considering the parameter savings. It is also noteworthy that on SNLI, DNLI, and SciTail, all the R2D2 -LSTM variants outperform the base LSTM model. Overall, we think that the parameter reuse properties, in addition to learning to share such reused parameter blocks amongst recurrent gating functions, may contribute to a regularizing effect.

## 4.2 NEURAL MACHINE TRANSLATION

Machine translation (MT) is concerned with translating between source-target language pairs. To this end, sequence transduction models are central to this problem domain. In this experiment, the key goal is to compare R2D2 Transformers against the standard Transformer model.

Table 2: Experimental results on neural machine translation (BLEU scores). $\dagger$ represents up-sampling with a factor of 2. R2D2 Transformers do not lose much performance despite enjoying huge saving in parameterization cost. Re-scaling to equal parameters can lead to improvement in results. Savings do not account for token embedding parameters.

| Setting | #Params | En-Vi | En-Id | De-En | Ro-En | En-Et | En-Mk | En-Ro |
|---|---|---|---|---|---|---|---|---|
| Transformer | 44M | 28.43 | 47.40 | **36.68** | **34.60** | 14.17 | 13.96 | **22.79** |
| Factorized | 11M (-75.0%) | 27.21 | 29.72 | 22.93 | 20.23 | 6.70 | 9.36 | 15.55 |
| Quaternion | 11M (-75.0%) | 28.00 | 42.22 | 32.83 | 30.53 | 13.10 | 13.67 | 18.50 |
| R2D2 $n = 2$ | 22M (-50.0%) | 29.25 | 46.32 | 35.52 | 33.40 | **14.98** | 13.60 | 21.73 |
| R2D2 $n = 4$ | 11M (-75.0%) | 29.13 | 44.13 | 35.53 | 32.74 | 14.11 | 13.01 | 21.19 |
| R2D2 $n = 8$ | 5.5M (-87.5%) | 29.34 | 40.81 | 34.16 | 31.88 | 13.08 | 12.95 | 21.66 |
| R2D2 $n = 16$ | 2.9M (-93.4%) | 29.04 | 33.48 | 33.89 | 31.53 | 12.15 | 11.97 | 19.63 |
| R2D2$^\dagger$ $n = 2$ | 44M | **29.54** | **49.05** | 34.32 | 33.88 | 14.05 | **14.41** | 22.18 |
| R2D2$^\dagger$ $n = 4$ | 22M (-50.0%) | 29.17 | 46.24 | 34.86 | 33.80 | 14.43 | 13.78 | 21.91 |
| R2D2$^\dagger$ $n = 8$ | 11M (-75.0%) | 29.47 | 43.49 | 34.71 | 32.59 | 13.75 | 13.78 | 21.43 |

Table 3: Experimental results on neural machine translation (BLEU scores) on WMT'16 En-De.

| Reported Models | |
|---|---|
| Transformer (Vaswani et al., 2017) | 27.30 |
| DynamicConv (Wu et al., 2019) | 29.70 |
| Transformer (our run) | 27.86 |
| **Compressed Models** | |
| Quaternion (Tay et al., 2019) | 25.14 |
| R2D2 Transformer ($n = 2$) | 26.43 |
| R2D2 Transformer ($n = 4$) | 26.52 |
| R2D2 Transformer ($n = 8$) | 24.11 |
| R2D2 Transformer ($n = 16$) | 23.01 |

Table 4: Training time (seconds per 100 steps) and Decoding Time (seconds to decode test set) with Beam size of 4 and length penalty of 0.6 on IWSLT 2014 German-English.

| Model | Train | Decoding |
|---|---|---|
| Vanilla | 7.79 | 341 |
| Factorized | 7.26 | 291 |
| Quaternion | 8.31 | 297 |
| R2D2 ($n = 4$) | 8.09 | 303 |
| R2D2 ($n = 8$) | 7.89 | 287 |

**Datasets and Setup**  We run experiments on 8 NMT datasets. The datasets are (1) IWSLT'15 English-Vietnamese (En-Vi)$^\dagger$, (2) IWSLT'17 English-Indonesian (En-Id)$^\dagger$, (3) IWSLT'14 German-English (En-De)$^\dagger$, (4) IWSLT'14 Romanian-English (Ro-En)$^\dagger$, (5) WMT'18 English-Estonian (En-Et)*, (6) Setimes English-Macedonian (En-Mk)*, (7) WMT'18 English-Romanian (En-Ro)* and (8) WMT'16 English-German (En-De).

Datasets with $^\dagger$ are run with 50K steps while datasets with $^*$ are trained for 100K steps also on a single GPU. For all tasks, we specify that Transformers have 4 layers, 8 heads, and a hidden size 512. We use beam size of 5 and $\alpha = 0.6$ (length penalty) for evaluating all the models. For all R2D2 models, we benchmark several settings for the hyperparameter $n = \{2, 4, 8, 16\}$. Likewise, we also re-scale the parameters by increasing the dimensions to match the original number of parameters. This is denoted by hyperparameter $U$. In the experiments, the upsampling factor is fixed at 2.

We benchmark against two baselines that are also concerned with reducing parameterization of the FC layer. The first, is the Factorized Transformer which approximates each FC layer with low-rank approximation. In this case, the number of latent factors is set to $\frac{d}{4}$ to enable direct comparison to the Quaternion Transformer. The second is the Hamilton product based Transformer proposed in (Tay et al., 2019).

**Experimental Results**   Table 3 reports our results on machine translation. Across 5 out of 7 NMT benchmarks, R2D2 Transformer outperforms the standard Transformer model. Overall, the empirical results are promising. On one hand, we observe that increasing $n$ all the way to 16 does not cause significant degradation in performance on datasets such as En-Vi. On the other hand, for most datasets, even with significant parameter reduction, we find that the decrease in the BLEU score is overall manageable ($\approx$ 1–3 BLEU points). However, we also note a rare occurrence where $n = 16$ results in significant decrease in the BLEU score, such as on the En-Id dataset.

On several datasets, the R2D2 Transformer model improves the performance. For example, datasets such as En-Vi and En-Et enjoy a performance boost of about 1 BLEU point with $n = 2$. Generally, there is only marginal performance degradation at $n = 4$ or $n = 8$ for most datasets. Finally, by upsampling with hyperparameter $U$, we are able to improve the performance of three datasets: En-Vi, En-Id, and En-Mk.

Additionally, we note that Factorized Transformer performs substantially worse than the vanilla Transformer or our R2D2-Transformer. On the other hand, our proposed R2D2 also makes reasonable gains over the Quaternion Transformer (Tay et al., 2019), signifying that flexible approximation of Hamilton products are effective.

Table 4 reports the decoding and training time for each Transformer variant. We observe that R2D2 with $n = 8$ has the fastest decoding speed amongst all variants. All in all, the training speed is also approximately similar. As such, this ascertains that our proposed dynamic weight diffusion method is not computationally expensive, despite its complexity.

## 4.3   TEXT STYLE TRANSFER

We experiment with sequence transduction for text style transfer. The goal of this task is to convert text of a certain style to another style.

**Datasets and Setup**   We use the Modern→Shakespeare corpus[2] in the experiments. The key goal here is to convert modern writing into Shakespeare writing. This dataset comprises of 18395 parallel sentences for training, 1218 parallel sentences for evaluation (dev set), and 1462 parallel sentences for testing. We still specify that Transformers have 4 layers, 8 heads, and a hidden size 512. Similar to NMT, we experiment with $n = \{2, 4, 8, 16\}$. We train all the models for 10K steps.

Table 5: Experimental results on text style transfer with Transformer models.

| Model | #Params | BLEU |
|---|---|---|
| Transformer | 44M | 11.65 |
| R2D2 ($n = 2$) | 22M (-50.0%) | 12.20 |
| R2D2 ($n = 4$) | 11M (-75.0%) | **12.42** |
| R2D2 ($n = 8$) | 5.5M (-87.5%) | 11.66 |
| R2D2 ($n = 16$) | 2.9M (-93.4%) | 10.76 |

**Experimental Results**   Table 5 reports the results on text style transfer. We observe that the best performance is achieved with R2D2 Transformer ($n = 4$). Notably, all except the $n = 16$ variant outperforms the standard Transformer model. This ascertains the effectiveness of the proposed R2D2 layer. This not only enables parameter savings but improves the performance of Transformer.

---

[2] https://github.com/tlatkowski/st

### 4.4 SUBJECT VERB AGREEMENT

We conduct additional experiments on subject-verb agreement task (Linzen et al., 2016). The task is a binary classification problem. It predicts if the sentence, *e.g.*, *'The keys to the cabinet ____.'* follows by a plural or a singular.

**Dataset and Setup**   This dataset can be found online (Linzen et al., 2016). We evaluate the R2D2 Transformer encoder against the standard Transformer encoder for the subject verb agreement task. In contrast to the previous experimental settings, we use a smaller Transformer architecture. Specifically, Transformers here have 2 layers, 4 head, and a hidden size 128. Since the hidden size is smaller than those in the previous experimental settings, we experiment with $n = \{2, 4, 8\}$.

Table 6: Experimental results on subject verb agreement (SVA) with Transformer models.

| Model | #Params | Acc |
|---|---|---|
| Transformer | 400K | 94.80 |
| Quaternion | 100K | 94.70 |
| R2D2 ($n = 2$) | 200K (-50.0%) | 95.14 |
| R2D2 ($n = 4$) | 101K (-74.8%) | 95.05 |
| R2D2 ($n = 8$) | 56K (-86.0%) | **95.62** |

**Experimental Results**   Table 6 reports the results on the SVA task. Results are promising, demonstrating that all variants with R2D2 layers outperform the standard Transformer. The best performance peaks at $n = 8$, despite parameter saving to up to $1/8$. R2D2 also outperforms Quaternion Transformers.

## 5 RELATED WORK

FC layers are ubiquitous in deep learning research (Goodfellow et al., 2016). Many, if not all neural building blocks incorporate this functionality, *i.e.*, transforming input vectors to an output space using a parameterized linear transformation matrix. Recurrent neural networks (Hochreiter & Schmidhuber, 1997), multilayer perceptrons, attention (Bahdanau et al., 2014), and Transformer models (Vaswani et al., 2017) are all heavily grounded in these transformation operations. Notably, recent state-of-the-art models in language domains aggressively exploit transformation layers, achieving very exceptional results (Devlin et al., 2018; Radford et al., 2019).

Our work can be interpreted as a form of soft parameter sharing, albeit dynamic and learned. A recent work (Savarese & Maire, 2019) proposed soft-weight sharing across stacked convolution layers, effectively simulating recurrence, *i.e.*, RNNs can be considered to be weight sharing across time. Quaternion networks (Zhu et al., 2018; Parcollet et al., 2018; 2019) are also known to inhibit weight sharing qualities and have demonstrated reasonable success despite having fewer parameters.

Low-rank approximations (Markovsky, 2012) and factorization methods are known techniques to reduce parameters and improve generalization. Majority of techniques aimed at reducing parameters of neural networks belong to this class (Sainath et al., 2013; Chen et al., 2018; Si et al., 2017). Within the context of language, GroupReduce (Chen et al., 2018) applies low-rank approximations to compress neural language models. (Si et al., 2017) proposed memory efficient kernels using low-rank approximation. Block diagonal approaches (Xie et al., 2017), structured sparsity (Gale et al., 2019) are also highly notable alternatives for improving the FC layer. Methods such as knowledge distillation (Hinton et al., 2015; Freitag et al., 2017) are also capable of reducing model complexity and can be considered an orthogonal direction from ours. In a similar vein, sharing parameters by the means of a meta network (HyperNetworks Ha et al. (2016) also perform a form of weight sharing.

## 6 CONCLUSION

We propose R2D2 layers, which can reduce the parameterization cost of their FC counterparts without significantly compromising performance. The R2D2 layer also offers a way to bridging multiplication between real space and hypercomplex space by subsuming and generalizing both matrix multiplication in real space and Hamilton product in hypercomplex space. It is highly modular and applicable to many dominant models such as LSTMs and Transformers. We evaluate such models equipped by R2D2 layers on comprehensive tasks, demonstrating substantial parameter savings with minimal degradation or improvement in performance.

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
