# OpenReview forum: "R2D2: Reuse & Reduce via Dynamic Weight Diffusion for Training Efficient NLP Models"
_ICLR.cc/2020/Conference — Reject_

### Official Review · AnonReviewer2 · 2019-10-18
**Official Blind Review #2**

**Rating:** 3

**Review:**

The authors propose R2D2 layers, which are trained to reduce and re-use existing parameters of a neural network layer, and apply this to Transformer and LSTM architectures. The authors conduct experiments on various NLP tasks, including NLI and NMT.

The main benefit of the proposed R2D2 layer is that the number of parameters can be reduced, as the existing parameters can be reused. I find this motivation compelling, particularly as it is well known Transformer networks are largely overparameterized.

Comments:
1. There is no analysis on the specific choices made for dynamic weight diffusion- the way the partitioning is done could have a large effect on the end result. There's also little comparison to other ways to share weights across a model besides the proposed weight diffusion method.

2. Sharing parameters contributes a regularization effect - it is difficult to untie the contributions of increased regularization from the proposed method. This is particularly problematic as the majority of the datasets used are "small" by current standards. WMT en-de (authors do not include the sizes of the datasets, but this is 4.5 million sentences) is the only large scale dataset, and the BLEU drop is quite large on this dataset compared to the smaller ones such as IWSLT.

To tie my points #1 and #2 together, I feel the authors did experiments on a variety of different tasks, but these style transfer and subject verb agreement tasks are not particularly interesting or realistic - instead this space should be devoted to discussions of the advantages of their method and analysis on its performance, which is quite lightly covered.

3. The authors claim that the R2D2 Transformer outperforms standard Transformer models on 5 out of 7 NMT tasks. This appears true if up-sampling with a factor of 2 is used to make the models larger again. The authors should compare to factorized/quaternion baselines which have a larger quantity of parameters as well.

4. Table 3, where results are reported on the competitive WMT en-de benchmark, lacks comparison for number of parameters and decoding speed. This table would probably have the most compelling and impactful results for this paper as this is the most competitive task (aside from the pre-training regime on MNLI/QNLI as part of GLUE). Can the authors complete this table so readers can understand the parameter reduction and inference speed possible from this method on this benchmark?

(As an aside, the technique should be applicable to the DynamicConv model, which is a Transformer variant?)

5. The related work section is quite light on other approaches to reducing model size, such as knowledge distillation or quantization? While the approach taken in this paper leverages parameter sharing, the motivation is similar and I feel acknowledging this entire area of work would be relevant.

6. I'm not clear on why we see inference time decoding speed improvements based on the description of the method. Can the authors clarify this point for me?

**Experience Assessment:**

I have published in this field for several years.

**Review Assessment: Checking Correctness Of Derivations And Theory:**

I assessed the sensibility of the derivations and theory.

**Review Assessment: Checking Correctness Of Experiments:**

I assessed the sensibility of the experiments.

**Review Assessment: Thoroughness In Paper Reading:**

I read the paper thoroughly.

---

> ### Author Response · Authors · 2019-11-15
> **Response**
>
> Dear Reviewer,
>
> Thanks for the insightful comments and feedback! We fully appreciate your time and effort in reviewing our work. We are also glad that you appreciate the motivation of our work.
>
> Regarding the choice of dynamic weight fusion, the key idea behind our approach is to dynamically learn the partitioning. Aside from the hyperparameter N, there are no straightforward choices of varying the partitioning (other than making them dynamic-sized which is a way more complex approach). The obvious partitioning strategy is the Quaternion method, which we make extensive comparisons with.
>
> Regarding regularization, we fully agree that decoupling effects of regularization and parameter savings is tricky. We believe that that this is interesting and warrants further investigation.
>
> Regarding the choice of experiments, machine translation and NLI are more “well-established” tasks. On the other hand, while the other 2 tasks are less popular, we decided to conduct these extra experiments to improve the diversity and coverage of the experiments. We agree that further analysis can help improve the paper and we are working on it for the updated version of the paper.
>
> Regarding decoding speed and parameter savings, the parameter size of Transformer base and R2D2 transformer base remains identical for all tasks (subject to only the vocab size). Moreover, the decoding speed, intuitively, should remain proportionate across different tasks.
>
> Regarding the related work, we have modified the paper with some references (as pointed out by Reviewer #3).
>
> Regarding decoding speed, we believe the reduced parameter size contributes to the improvement in speed. In particular, the matrix multiplication operations in R2D2 networks are now smaller.
>
> Once again, thank you for taking the time to review our work!

---

### Official Review · AnonReviewer1 · 2019-10-23
**Official Blind Review #1**

**Rating:** 6

**Review:**

This paper proposes a new Reuse and Reduce with Dynamic weight Diffusion (R2D2) layer as an alternative to feed-forward layers in neural networks. The layer is inspired by the Hamilton Product in a hypercomplex space (where numbers have multiple imaginary components). The main idea is to have two smaller parameter blocks that are partitioned, multiplied, and concatenated together in order to form the full weight matrix of the feed-forward layer.
In extensive experiments on NLI, NMT, text style transfer and subject-verb agreement, feed-forward layers in LSTMs and Transformers are replaced with R2D2 layers. The modified models achieve similar performance to the originals, while being more than 50% smaller.

Overall, the proposed method is presented clearly and the experiments are comprehensive and convincing. For these reasons, I am leaning towards accepting this paper.

The proposed method is well explained. In particular, Figure 1 is helpful to obtain a conceptual picture of the method. This is in contrast to some of the previous methods based on hypercomplex operations, which often seem harder to grasp and visualize. In addition, it is helpful that connections to other operations such as matrix multiplication and the Hamilton product are highlighted.

The proposed method is evaluated extensively. It is applied to different models (LSTMs and Transformers) and on different tasks. Results are mostly convincing, as performance numbers are competitive with the baselines, while the models are much smaller. In addition, it compares to previous work, which it outperforms.

The main thing that I'm missing is some analysis of the dynamics of the model, what it is learning (in comparison to using FC layers) or why a smaller number of parameters is still competitive with the standard FC layers. Are feed-forward layers over-parameterized and only a smaller number of their weights are actually used in practice, similar to lottery tickets (https://arxiv.org/abs/1803.03635)? How do the learned A and S blocks look like? Is the entire model learning a different function or do the R2D2 layers just find a way to approximate a feed-forward layer?

Overall, as the method seems straightforward enough to implement and achieves promising results, it has the potential to have some practical impact.

**Experience Assessment:**

I have published one or two papers in this area.

**Review Assessment: Checking Correctness Of Derivations And Theory:**

I assessed the sensibility of the derivations and theory.

**Review Assessment: Checking Correctness Of Experiments:**

I carefully checked the experiments.

**Review Assessment: Thoroughness In Paper Reading:**

I read the paper thoroughly.

---

> ### Author Response · Authors · 2019-11-15
> **Response**
>
> Dear Reviewer,
>
> Thanks for the insightful review! We are happy to hear that you liked our paper!
>
> Pertaining to the dynamics of the model, we believe that our method is a more expressive, parameterized adaptation of the Hamilton product - which already brings about benefits from latent inter component interactions. We think that this helps the model learn to approximate an actual FC layer with less parameters, although not completely. Additionally, there are also motivation factors such as regularization from weight sharing. We will be continuing on this line of work to investigate this carefully and are thankful for the points you have brought up. Updated appendices of the visualized blocks (A and S) will be updated by the next version of the paper.
>
> Once again, thanks for taking the time to review our paper and we are happy about the positive review.

---

### Official Review · AnonReviewer3 · 2019-10-24
**Official Blind Review #3**

**Rating:** 6

**Review:**

Paper Summary:

This paper proposes to train smaller models by decomposing the weights of fully connected networks as the product of smaller matrices, along with a reordering/transpose of the outcome. The experiments shows that models with less parameters yield comparable performance with their larger counterparts.

Review Summary:

The method is technically sound and the paper reads well. Experiments demonstrate the efficacy of the method, although some ablations are missing (see below). The paper is however not clear on the ultimate objective of the method (speed/accuracy/generalisation?) and does not compare with alternatives.

Detailed Review:

The introduction does not make clear if your motivation to make model smaller is training speed, inference speed, memory usage, generalization accuracy. Please clarify.

The explanation of the method, i.e. Section 2.2.1, is not clear, in particular for the mapping \psi. I feel it would cleared if somewhere in the paper there was an equation with the element-wise correspondence, i.e. H_{?,?} = \sum_k A_i,k S_k,j
In that section, you should introduce that n is a hyperparameter before using it as well.
In that section, you could also discuss parameter initialization, and whether this model can use weight decay over H or A/S. it is also not clear to me if you control the norm ratio between A and S given the weight magnitude is over parameterized.

The experimental section lack a validation/ablation study to help the reader understand the interplay between the number of blocks and the number of latent dimensions. It will also be good to show learning curves to compare training speed of different parameterization.
Also no training errors are reported, does your method can be seen as a regularizer, i.e. is training objective closer to valid objective when n grows? Did you have to change other regularization parameters like dropout.

To me the main weakness of the paper lies in the lack of comparison with alternatives. Replacing fully connected layers with alternative has a rich literature that the authors ignore.
I feel it is necessary to compare the approach with
(i) block diagonal approaches, popular since ResNext for convolutions but equally applicable to linear layers. https://arxiv.org/abs/1611.05431
(ii) other form of structured sparsity. https://arxiv.org/abs/1902.09574 (survey). https://arxiv.org/abs/1812.08301 (squantizer) https://arxiv.org/abs/1802.08435 (block sparsity)...
(iii) distillation of large models into smaller models.  https://arxiv.org/abs/1503.02531 https://arxiv.org/abs/1702.01802
(iv) it might not be necessary to compare, but at least mentioning approaches which predict weights from a meta network would be good.  https://arxiv.org/abs/1609.09106

As a reviewer, I am a bit annoyed that made no effort to have a decent list of related work and that they delegate that work to the reviewers to do so.

Details:
"transformation layer": this is not common terminology, please prefer linear layer or fully-connected layer.
please define all acronyms, e.g. FC.
The experimental section does not define \alpha (end of page 6).


**Experience Assessment:**

I have published in this field for several years.

**Review Assessment: Checking Correctness Of Derivations And Theory:**

I carefully checked the derivations and theory.

**Review Assessment: Checking Correctness Of Experiments:**

I carefully checked the experiments.

**Review Assessment: Thoroughness In Paper Reading:**

I read the paper thoroughly.

---

> ### Author Response · Authors · 2019-11-15
> **Response**
>
> Dear Reviewer,
>
> Thanks for the insightful comments and feedback, along with spending valuable time to review our paper!
>
> In lieu of your detailed feedback, we have made the following changes to the paper.
> 1) Made it clearer that the proposed method is concerned with improving the memory footprint (parameter complexity).
> 2) Included a discussion and citation of the above-mentioned suggested literature (block diagonal, structured sparsity etc) to improve the completeness of the related work. To this end, we believe a detailed empirical comparison of all "memory-saving" methods is indeed interesting future work. We feel that this extended analysis is better dedicated to a follow-up work to comprehensively study the effect of orthogonal methods such as sparsity, quantization, low-precision, distillation etc. This is also partly in lieu of the limited window of the response period and we feel it would be better to carefully evaluate these methods without having a tight time constraint. The factorized and quaternion baselines are the closest to our method which we have selected to focus on in this paper.
> 3) Introduced to the reader that N is a hyperparameter.
> 4) Improved the clarity of experimental settings by stating that all parameter initialization and hyperparamters remains identical to the vanilla version.
> 5) Defined alpha (length penalty)
>
> We fully agree that understanding the regularizing effect of our method is key. At this moment, we do not that any quantifiable results on hand. However, we can offer some first hand insights that the training curves (loss and val) of R2D2 Transformer are similar to the Vanilla Transformer. This is based on experience from developing this method.
>
> Pertaining to the different block sizes, our tabular results do offer an analysis of different values of n (i.e., extents of memory savings).
>
> Once again, thanks for taking the time to review our work!

---

### Decision · Program_Chairs · 2019-12-19

**Decision:**

Reject

**Comment:**

This paper proposes a very interesting alternative to feed-forward network layers, based on Quaternion methods and Hamilton products, which has the benefit of reducing the number of parameters in the neural network (more than 50% smaller) without sacrificing performance. They conducted extensive experiments on language tasks (NMT and NLI, among others) using transformers and LSTMs.

The paper appears to be clearly presented and have extensive results on a variety of tasks. However all reviewers pointed out that there is a lack of in-depth analysis and thus insight into why this approach works, as well as questions on the specific effects of regularization. These concerns were not addressed in the rebuttal period, instead leaving it to future work. My assessment is that, with further analysis, ablation studies, and comparison to alternative methods for reducing model size (quantization, etc), this paper has the potential to be quite impactful, and I look forward to future versions of this work. As it currently stands, however, I don’t believe it’s suitable for publication at ICLR.